# Preparation and characterization of monoclonal antibodies recognizing two CD4 isotypes of Microminipigs

Shino Ohshima[1], Tatsuya Matsubara[2], Asuka Miyamoto[1], Atsuko Shigenari[1], Noriaki Imaeda[2], Masaki Takasu[2], Masafumi Tanaka[1], Takashi Shiina[1], Shingo Suzuki[1], Noriaki Hirayama[3], Hitoshi Kitagawa[4], Jerzy K. Kulski[1,5], Asako Ando[1], Yoshie Kametani[1,3]*

1 Division of Basic Medical Science, Department of Molecular Life Science, Tokai University School of Medicine, Isehara, Kanagawa, Japan, 2 Department of Veterinary Medicine, Faculty of Applied Biological Sciences, Gifu University, Gifu, Gifu, Japan, 3 Institute of Advanced Biosciences, Tokai University, Hiratsuka, Kanagawa, Japan, 4 Department of Veterinary Medicine, Faculty of Veterinary Medicine Okayama University of Science, Imabari, Ehime, Japan, 5 Faculty of Health and Medical Sciences, UWA Medical School, The University of Western Australia, Crawley, WA, Australia

* y-kametn@is.icc.u-tokai.ac.jp

**Data Availability Statement:** All relevant data are within the manuscript and its Supporting Information files.

## Abstract

Cluster of differentiation 4 (CD4) molecule expressed on the leukocytes is known to function as a co-receptor for class II major histocompatibility complex (MHC) binding to T cell receptor (TCR) on helper T cells. We previously identified two CD4 alleles (CD4.A and CD4.B) in a Microminipig population based on nucleotide sequencing and PCR detection of their gene sequences. However, CD4.B protein expression was not examined because of the unavailability of a reactive antibody to a CD4.B epitope. In this study, we have produced two swine-specific monoclonal antibodies (mAbs) against CD4.B molecules, one that recognizes only CD4.B (b1D7) and the other that recognizes both the CD4.A and CD4.B alleles (x1E10) and that can be used to distinguish CD4 T cell subsets by flow cytometry and immunohistochemistry. Using these two mAbs, we identified CD4.A and CD4.B allele-specific proteins on the surface of CD4.A (+/+) and CD4.B (+/+) T cells at a similar level of expression. Moreover, stimulation of peripheral blood mononuclear cells (PBMCs) derived from CD4.A (+/+) and CD4.B (+/+) swine with toxic shock syndrome toxin-1 (TSST-1) *in vitro* similarly activated both groups of cells that exhibited a slight increase in the CD4/CD8 double positive (DP) cell ratio. A large portion of the DP cells from the allelic CD4.A (+/+) and CD4.B (+/+) groups enhanced the total CD4 and class I swine leukocyte antigen (SLA) expression. The x1E10 mAb delayed and reduced the TSST-1-induced activation of CD4 T cells. Thus, CD4.B appears to be a functional protein whose expression on activated T cells is analogous to CD4.A.

**Funding:** The authors received no specific funding for this work.

**Competing interests:** Thee authors have declared that no competing interests exist.

## Introduction

Cluster of differentiation 4 (CD4), a glycoprotein expressed on immune competent cells such as lymphocytes, monocytes, macrophages and dendritic cells, is widely used as a marker of T cell subsets for the functional analysis of the immune response. CD4 possesses four extracellular immunoglobulin-A subdomains (D1 to D4) [1, 2] and an intracellular tyrosine kinase Lck binding domain that enhance TCR signals [3, 4]. The extracellular domain of CD4 is thought to function as a coreceptor of TCR-MHC cognate interaction [5]. The extracellular domain contains a subdomain called the D1 domain, which binds to a non-polymorphic class II MHC β2 domain [1]. However, there are reports that CD4 also binds directly to the class II MHC molecule with low affinity in a TCR independent manner [6, 7] suggesting that CD4 might function as an adhesion molecule to induce signal transduction [8].

The CD4 receptor is polymorphic in a number of different species [9–13]. In humans, CD4 polymorphisms were correlated to some infectious diseases such as HIV infection [14] and chronic diseases such as type-1 diabetes [15]. Although Microminipigs and NIH minipigs have at least two CD4 isotypes, they could not be distinguished by conventional anti-CD4 antibodies [16, 17]. By sequencing the Microminipig CD4 variant genes, we found that a CD4 allele named CD4.B had 10 amino-acid substitutions in the CD4 domain 1 region in comparison with the swine CD4 amino-acid reference sequence [Gen-Bank: NP_001001908] [18], whereas CD4.B was not recognized by three conventional monoclonal antibodies (mAbs) named 74-12-4, MIL17, and PT90A [17]. The 10 amino-acid substitutions in CD4.2 NIH minipigs were also found in the same positions as those in CD4.B [18]. These regions of amino-acid substitution overlapped with the site of class II MHC recognition by CD4 indicating that the allelic polymorphism may affect the acquired immune response. The percentage frequency of the Microminipigs with the CD4.B allele, not recognized by the available mAbs, was at most 51.1%. No obvious difference of immune cell ratio or severe immunodeficiency was observed in the Microminipig herd [17]. Also, Blanc et al. reported that a high genetic variability of CD4 in the melanoblastoma-bearing Libechov minipig (MeLiM) strain was associated with the immune response against the skin cancer [9].

Microminipigs are novel, extra-small miniature pigs developed for biomedical research in Japan [19, 20]. Recently, we assigned different swine leukocyte antigen (SLA) class I and class II haplotypes to our herd of Microminipigs [21]. We found that the *in vitro* administration of toxic shock syndrome toxin-1 (TSST-1) activated polyclonal T cells and increased class I SLA transcription showing that TSST-1 and the SLA together are useful for evaluating the *in vitro* activation level of swine T cells [22, 23].

In this study, we developed two mAbs which recognize Microminipig CD4.B, one specific to CD4.B and the other to CD4.A and CD4.B in order to analyze CD4.A and CD4.B protein expression and function. We used these two antibodies to evaluate (1) the level of CD4 protein expression and T cell activation, and (2) the association of T cell activation with class I SLA protein expression levels in response to *in vitro* T cell stimulation with theTSST-1.

## Materials and methods

### Animals and tissues

Experiments using mice were approved by the Institutional Committee for Animal Care of Tokai University and were performed following the University guidelines (The approval numbers are #152030, #170426). Swine study was approved by the Animal Care and Use Committee of Gifu University (The approval number is #17042, May 26, 2017). Eight-week-old female BALB/c mice were purchased from CLEA Japan (Tokyo, Japan) and kept under specific

pathogen-free conditions. Adult swine peripheral blood was provided by Fuji Micra Inc and Gifu University. The sex, date of birth, CD4 type and MHC haplotypes of the 14 Microminipigs used in this study are presented in S1 Table. A marmoset blood sample was supplied from Central Institute of Experimental Animals (Kawasaki, Japan). A human blood sample was purchased from Cosmo Bio Co. Ltd (Tokyo Japan). This study was carried out according to the laboratory animal guidelines of Gifu University, Tokai University and Central Institute for Experimental Animals.

### SLA genotyping

Eight SLA haplotypes were assigned to a herd of Microminipigs with extremely small body sizes by nucleotide sequence determination of reverse transcriptase-polymerase chain reaction (RT-PCR) products and low-resolution SLA genotyping using PCR-sequence-specific primers (SSP) of the three classical class I SLA genes, *SLA-1*, *SLA-2* and *SLA-3*, and class II genes, *DRB1* and *DQB1* [21]. In this study, *SLA-1*, *SLA-2*, *SLA-3*, *DRB1* and *DQB1* alleles in Microminipigs were assigned by low-resolution SLA genotyping using the aforementioned PCR-SSP method as described previously [21, 24, 25]. The primers are shown in S2 Table. Eight types of SLA haplotypes were deduced from an analysis of the inheritance and segregation of alleles of the three class I and two class II SLA genes in descendants of the Microminipig population. PBMC of Microminipigs with two kinds of homozygous low-resolution class I and class II SLA haplotypes, Hp-35.23 and Hp-43.37, were used for comparison of class I SLA genes expression on T cells with different CD4 isotypes.

### CD4 genotyping

We assigned CD4 genotypes in the Microminipig herd using a PCR-RFLP method [18]. Briefly, PCR amplification was performed on genomic DNA to amplify CD4 exon 3 using primers shown in S3 Table, and the PCR products were digested with a restriction enzyme *Bse*RI (New England Biolabs Inc., Ipswich, MA). The digested samples were electrophoresed in 2% agarose gel to assign the genotypes.

### Transfection

The transfected cDNA sequences of the swine *CD4.A* and *CD4.B* genes were based on the *Sus scrofa CD4.A* and *CD4.B* mRNA sequences LC064059 and LC064060, respectively. The primers used to generate the cDNA sequences are shown in S3 Table. RNA was extracted from cells using TRIzol (Invitrogen, Carlsbad, CA, USA) and reverse-transcribed to cDNA using the High Capacity cDNA Reverse Transcription Kit (Life Technologies, Carlsbad, CA, USA). The modified S/MAR (scaffold/matrix attachment region) episomal vectors [26] carrying *CD4.A* or *CD4.B* were used for the transfection. Transgene-positive cell ratio was determined according to the fluorescent intensity of the co-expressed mVenus protein [27].

HEK293 cells, a cell line derived from human embryonic kidney cells, were cultured in D-MEM (GE Healthcare, Buckingham, UK)-10% fetal calf serum (FCS) (bisera, Kansas City, MO, USA) medium. A20 cells, a BALB/c B lymphoma cell line derived from a spontaneous reticulum cell neoplasm, were cultured in RPMI 1640 (Nissui Pharmaceutical co. Ltd. Tokyo, Japan)-10% FCS medium. The cells were transfected with the cDNA sequence inserted in the modified S/MAR episomal vector using the Invitrogen Neon transfection system (HEK293 cells: 1100 V, 10 ms, 3 pulses; A20 cells: 1500 V, 10 ms, 3 pulses) and cultured in 5% $CO_2$ at 37°C for 18 ~24 hrs added with G418 (Roche Diagnostics, Indianapolis, USA). Cultured cells were collected and the expression of the mVenus reporter gene was measured by flow cytometry (Becton Dickinson, NJ, USA).

## Preparation of the peripheral blood mononuclear cells (PBMCs)

Peripheral blood samples from Microminipigs (S1 Table) were collected into separate heparinized tubes and centrifuged on Lymphoprep (Axis-Shield, Oslo, Norway) at 670 x g for 30 min. The peripheral blood mononuclear cells (PBMCs) were collected and washed with 30 ml of 1% (w/v) bovine serum albumin (BSA) containing phosphate-buffered saline (PBS). The remaining erythrocytes were lysed osmotically. The PBMCs were washed with PBS and used for further experiments. The single human PBMC sample was purchased from Cosmo Bio Co. Ltd (Tokyo Japan), and the mouse and marmoset PBMCs were collected using Lymphoprep, as described above.

## Monoclonal antibody preparation

A summary of the protocol used for immunization of the BALB/c mice with swine CD4.A and CD4.B proteins prior to fusion of the mouse splenocytes with the mouse myeloma cell line is outlined in S1 Fig. We initially immunized BALB/c mice with mitomycin C (MMC, Kyowa-hakko-Kirin, Tokyo Japan)-treated swine CD4.A or CD4.B homozygous PBMCs (1 to 3 $\times 10^6$ cells/animal). For booster treatments, mice were injected biweekly with MMC-treated A20 transfectants ($CD4A^+$/A20 and / or $CD4B^+$/A20) for up to 5 times with $1 \times 10^6$ cells/animal. Prior to the injections, the MMC (final, 0.04 mg/ml) was added to the A20 transfectant cultures and incubated at 37°C for 30 min in 5% $CO_2$. MMC-treated A20 transfectants ($1 \times 10^6$) were used for the last booster. The mice were anesthetized with 20% isoflurane and the blood was collected from the orbit. The antibody reactivity of immunized mouse blood plasma was checked by flow cytometry using $CD4.A$ or $CD4.B$ cDNA-transfected HEK293 cells as a source of antigen. After 4 days of the final boost, mice were anesthetized with 20% isoflurane and sacrificed by blood removal; extracted splenocytes were fused with the mouse myeloma cell line P3-X63-Ag8-U1 (P3X) purchased from Riken (Saitama, Japan) according to a standard procedure with Polyethylene glycol (PEG) purchased from Merk Japan co. Ltd (Tokyo, Japan) [28]. RPMI 1640 medium and supplements were purchased from Nissui co. Ltd (Tokyo, Japan) (RPMI 1640), Sigma-Aldrich (HAT) and Thermofisher Scientific K.K. (Tokyo, Japan) (HT). Positive clones were identified by flow cytometry or using an Imaging Analyzer (Array Scan, Thermo scientific, MA, USA). Briefly, $CD4.A$ or $CD4.B$ transfected HEK293 cells were mixed with non-transfected HEK293 cells and plated in the wells of 96-well plates. Culture supernatants were added to each well, incubated for 15 min and washed twice. APC-labeled (APC: allophycocyanin) anti-mouse IgG polyclonal antibody (Poly4053; Bio Legend, San Diego, US) was added and incubated for 15 min. Plates were washed and stained with Hoechst 33342 (Invitrogen, Oregon, USA) for 30 min at room temperature and analyzed using the Imaging Analyzer. Positive cells were picked according to the fluorescence intensity of APC and the co-expressed mVenus fluorescent protein. The same analysis was repeated to select the stable positive clones that were then isolated, expanded and stocked. The mAb isotype was determined using a mouse monoclonal antibody isotyping kit (Iso Strip, Roche, Basel Schweiz). The sequence of each monoclonal antibody was determined by Repertoire Genesis Inc. (Osaka Japan).

## Flow cytometry

Cells were incubated with appropriately diluted, fluorescently-labeled primary mAb for 15 min at 4°C and washed with 1% (w/v) BSA-containing PBS. In some cases, cells were re-incubated with labeled secondary antibodies. The mAb x1E10 was biotinylated by Biotin Labeling Kit (Cosmo Bio co., Ltd, Tokyo Japan) according to manufacturer's protocol. The mAbs that were used in our analyses are shown in S4 Table. The cells were analyzed using FACS Verse

(BD Bioscience, Franklin Lakes, NJ). For each sample, the live gate with white blood cells or lymphocytes was analyzed by FlowJo (TOMY Digital Biology).

## Primary sequence and structure analysis of mAb

Total RNA was extracted from specific hybridomas, and the cDNA was checked for the amplification of immunoglobulin heavy- and light-chain-specific genes. Sequence reactions were performed with GenomeLab DTCS Quick Start Kit (Beckman Coulter, Brea, CA, USA) and analyzed using a CEQ8000 Genetic Analysis System (Beckman Coulter). The software Genetyx (GENETYX co. Ltd, Tokyo, Japan) was used for sequence prediction.

## Model building of the 3D structure of CD4 molecules

The 3D models of the CD4.A and CD4.B molecules translated from sequenced genes [18] were constructed using the homology modeling method [29] implemented in the software system Molecular Operating Environment (MOE) (2019.01; Chemical Computing Group ULC, 1010 Sherbrooke St. West, Suite #910, Montreal, QC, Canada, H3A 2R7, 2019). The protein structure (PDB ID, 3JWD) with the highest sequence identity to CD4.A and CD4.B was selected from the Protein Data Bank(PDB) [30] and used as a template for homology modeling. The amino acid sequence of CD4.A and CD4.B are compared in Fig 1.

## Histochemical analysis

Swine frozen tissues were micro-sectioned (5 μm) and fixed with 4% buffered paraformaldehyde for 20 min (Wako Pure Chemical Industries, Ltd), then washed and endogenous peroxidase was blocked for 10 min at room temperature. The sections were blocked with goat serum for 30 min, washed and then the primary monoclonal antibody was added. Subsequent incubation with peroxidase-labeled anti-mouse Ig was performed according to the manufacturer's protocol. The list of antibodies are shown in the S4 Table.

## Stimulation of PBMCs

Swine peripheral blood samples were collected into heparinized tubes and centrifuged on Lymphocepar (IBL Co. Fujioka, Japan) at 670 x g for 30 min. PBMCs were isolated and washed with 30 ml of 1% (w/v) BSA-containing PBS (PBSA) by centrifuging at 350 x g for 5 min. The remaining erythrocytes were lysed osmotically. The PBMCs were washed and cultured ($3x10^6$/ well) in RPMI 1640 medium containing 10% FCS in the presence of the TSST-1 (Toxin Tec. Sarasota, USA) at 1 μg/mL at 37°C and 5% $CO_2$. After 72 hrs, the cells were collected, washed with PBS andanalyzed by flow cytometry (FCM). The TSST-1/mAb inhibition assay was performed by adding the purified mAb (x1E10) to the culture medium with TSST-1 at 1 μg/mL. The cells were analyzed at 24, 48 and 72 hrs to detect the surface expression of CD4, CD8 and class I SLA by FCM.

## Statistical analysis

Regression lines and regression equation for age and CD4+ cell correlation analysis, and the student's *t*-test for the comparison of class I SLA expression on the TSST-1-stimulated and non-stimulated cells were performed by Microsoft EXCEL (Microsoft Office 2016, Microsoft Corporation, Redmond, WA).

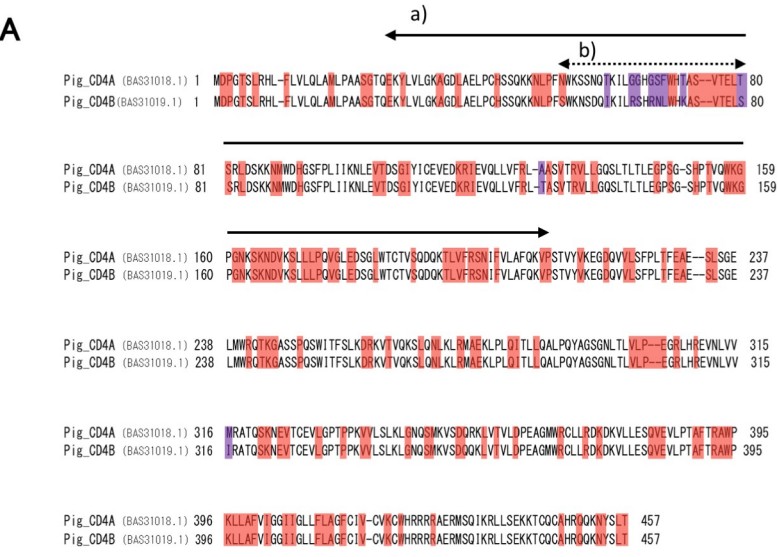

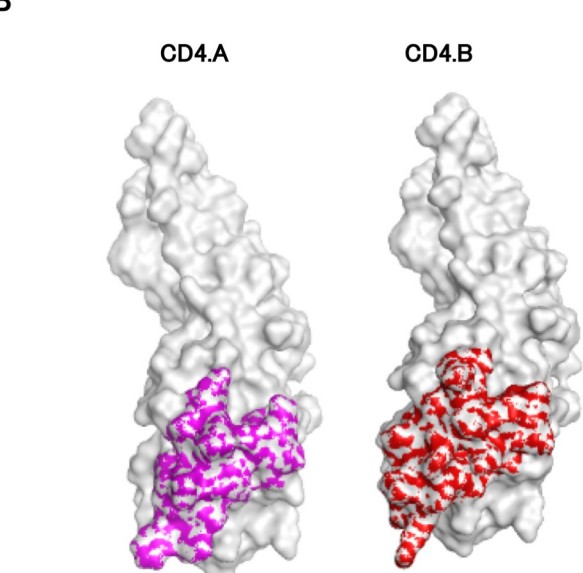

**Fig 1. Microminipig CD4 protein structure.** A. Amino acid sequence alignments of two alleles of the CD4 locus (CD4.A, upper line; CD4.B, lower line) [18]. Amino acid substitutions between CD4.A and CD4.B are shown in purple color. Identical amino acid sequences between CD4.A and CD4.B but different from mouse and human are shown in red color. Other sequences indicate identical amino acids among human, mouse and swine. a) the region N;27-N;210 aa corresponding to the C chain of the reference tertiary structure 3JWD obtained from PDB [32], b) the predicted epitope sites for recognizing CD4.A and CD4.B. B. The tertiary structures of the CD4.A (left panel), and CD4.B (right panel) N;27-N;210 aa constructed based on the 3D structure of 3JWD [32]. Purple (CD4.A) and red (CD4.B) colors represent the amino acid sequences (N;54-N;80) that are predicted to be different epitope-candidate sites. As the molecular surfaces of the 3D models are significantly different, their antibody binding affinities should be different.

## Results

### Simulation of swine CD4 isotype epitope

Because CD4 possesses four Ig domains in the extracellular domain, we first analyzed the similarity of CD4.A and CD4.B amino acid sequences (Fig 1A) archived at NCBI [18]. In a

previous study we found that the CD4.A protein was recognized by available mAbs, whereas the CD4.B protein was not recognized by these mAbs [17]. Based on the primary amino acid sequences of CD4.A and CD4.B [18], we predicted the spatial position of the diverged sequence and the putative epitope sites (Fig 1B). The human CD4 was predicted to bind to class II MHC at the site N;99–226 [31]. On the other hand, the diverged amino acid sequences in CD4.A and CD4.B of Microminpigs were located on N;54–80, which is a little upstream of the predicted sites for human CD4 and class II MHC binding. However, there are two amino acid variants also in the MHC-CD4 binding region (Fig 1A) that might influence TCR-class II MHC interactions. Therefore, we concluded that it would be necessary to prepare specific mAbs against the CD4 isotypes in order to identify and compare the activated CD4.A and CD4.B T cells and confirm whether these T cells can be activated similarly or differently by TCR stimulation.

To examine whether the diverged sequence within the swine CD4 might contain a predicted and unique epitope for recognition by a mAb, we selected a human CD4 reference sequence, which interacts to HIV-1 gp120 (PDB ID: 3JWD) [32, 33] as the protein with the greatest amino acid similarity to the porcine CD4 (<70% identity) for a sequence comparison of possible epitope motifs. As shown in Fig 1A (a), the identities between the human C chain of the protein reference sequence 3JWD (CD4 region) and the 27–210 amino acid sequences of the swine CD4.A or CD4. B chains were 56.5% and 55.4%, respectively. The tertiary structure corresponding to the N-terminal 26 residues and the residues after 211th residue were not present in the 3JWD C chain. Therefore, the tertiary structure of the N;27 -N;210 amino acid residue domain (N-domain) was constructed by homology modeling [29] using the structure of 3JWD corresponding to the N-domain. We predicted the site of diverged amino acid sequence on the 3D model that was shown in Fig 1B. Each of the N;54-N;80 amino acids (predicted as different epitope sites) were colored in purple and red (Fig 1A) with putative epitope sites were exposed on the surface of both molecules (Fig 1B). Moreover, the five amino acid replacements (T61I, G65R, G66S, G68R and T73K) have a difference in polarity and charge and the tertiary structure of the two molecules is different in our homology modeling (Fig 1B). These molecular differences suggest that the antigenicity of the Microminipig CD4.A and CD4.B is modified by the sequence diversity and that each protein might have unique antibody epitopes that could be distinguished by mAbs.

## Preparation of mAbs recognizing swine CD4 isotypes

We prepared two specific mAbs, one that recognized only CD4.B and the other that recognized both CD4.A and CD4.B molecules. In the preliminary analyses, MMC-treated transfectants (*CD4A*⁺/A20 or *CD4B*⁺/A20) were injected multiple times to booster the immunization of BALB/c mice (S1A Fig). In the analyses, the reactivity of the antiserum to HEK293 cells expressing either CD4.A or CD4.B after the second booster (S1B Fig) was confirmed by flow cytometry. Therefore, the mice with high fluorescence intensity were selected and used for the monoclonal antibody preparation (S1B Fig).

BALB/c mice were immunized with CD4.B homo-type (CD4.B (+/+)) PBMC twice and boosted with *CD4.B* transfected A20 cells. In these mice, both the CD4.B and the CD4.A specific antibodies were produced (S1B Fig). The spleen cells of immunized mice with the highest antisera reactivity to both *CD4.A* and *CD4.B* gene products were selected and fused with the mouse myeloma cell line P3X. The supernatants of hybridomas were screened for staining only CD4.B or both CD4.A and CD4.B using *CD4.A* and *CD4.B* gene-transfected HEK293 cell lines with Array Scan.

The first clone (b1D7) that we obtained for further analysis recognized only the CD4.B protein and not the CD4.A protein (Fig 2A and 2B). This mAb did not react with human, mouse

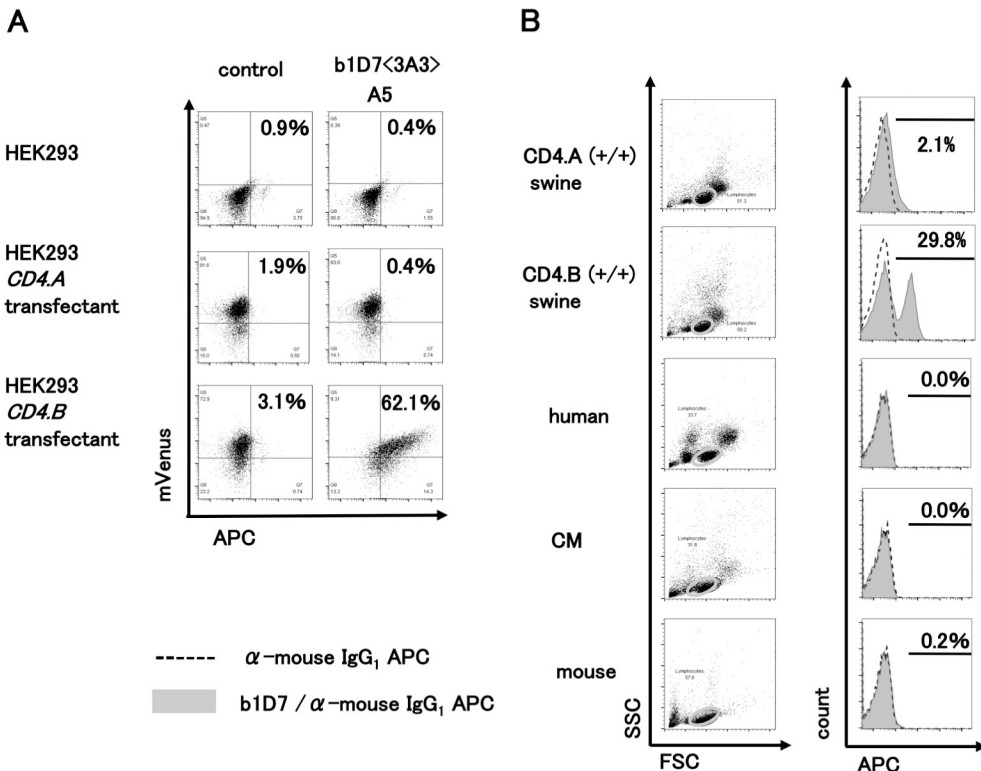

**Fig 2. Characterization of anti-swine CD4.B mAb (b1D7).** A. HEK293 and the transfectants of *CD4.A* or *CD4.B* were first stained with b1D7 (subclone number; 3A3-A5) clone supernatant, then stained with anti-mouse IgG-APC, and analyzed by FCM. The numbers in the panels show the percentages of mVenus and APC double-positive cells. B. Swine PBMCs (CD4.A (+/+) and CD4.B (+/+)) were stained with the purified mAb and anti-mouse IgG-APC, and analyzed by FCM. The species specificity was examined using CD4.A (+/+) and CD4.B (+/+) swine, human, CM and mouse PBMCs. Lymphoid gate (orbits) was used for the analysis (left panels). Shaded peaks represent the b1D7-stained patterns. Broken lines represent the isotype-control-stained patterns (right panels). The bars in the histograms indicate the presence of b1D7 positive cell fractions together with the percentage of b1D7 positive cells.

and common marmoset (CM) PBMCs, but it recognized swine PBMCs (Fig 2B) suggesting species specificity. The mRNA was extracted from the hybridoma and next-generation sequencing revealed that the antibody possessed heavy chain, IGHV3-2_x_IGHJ3_IGHG1_C ARQGYGDYDPFPYW, and two light chains IGKV6-20_IGKJ2_CGQGYSYPYTF and IGKV4-55_IGKJ2_CQQWSSYPYTF (S2 Fig). These light chains were constitutively expressed even after cloning suggesting that the two B cells were fused with one myeloma cell.

The second clone that we obtained for further analysis (x1E10) recognized both the CD4.A and the CD4.B proteins. The antibody secreted by the clone did not recognize human, mouse or common marmoset PBMCs, but reacted specifically with swine PBMCs (Fig 3A and 3B). Next generation sequencing of the mRNA extracted from the hybridoma showed that the antibody possessed a heavy chain, IGHV3-1_IGHD2-9 and IGHD2-4_IGHJ4_CARRTYYDY-DYYGMDYW_IGHG2A and a light chain, IGKV1-117_IGKJ1_CFQGSHVPWTF (S3 Fig). This antibody also reacted with swine splenocytes in an immunohistochemical analysis of CD4.A (+/+) or CD4.B (+/+) spleen tissue sections revealing weakly colored small cells scattered in periarteriolar lymphoid sheaths (PALS) and strongly colored larger cells scattered around the white pulp (Fig 3C). The strongly colored larger cells scattered around the white pulp in the negative-control suggests a non-specific binding reaction due to the secondary

antibodies. Collectively, these results confirmed the specificity and usefulness of the newly developed mAbs against CD4.A and CD4.B of swine T cells.

## Comparison of CD4 and CD8 expression in activation-induced T cells with different CD4 isotypes

To evaluate the utility of our new antibody that interacted with CD4.A and CD4.B, we first analyzed the expression of CD4 and CD8 molecules on the swine PBMCs using the anti-CD4AB mAb (x1E10) that we prepared in the present study and a commercially available anti-CD8 mAb, respectively (S4 Table). As shown in Fig 4A, four samples of CD4.A (+/+) and four samples of CD4.B (+/+) swine PBMCs showed similar patterns of CD4 and CD8 expression. In the swine PBMCs, not only CD4 single-positive (CD4SP) and CD8 single-positive (CD8SP) cells, but also the CD4/CD8 double-positive (DP) cells were detected. The range of total CD4 + T cell ratio was 21–42% for CD4.A (+/+) and 25–35% for CD4.B (+/+). We verified a previous report [34] that the younger swine PBMCs contained fewer DP cells, and we also found that the relationship between the age of swine and DP cell ratio was positively correlated ($r^2$ = 0.6083 for CD4.A (+/+) and $r^2$ = 0.7474 for CD4.B (+/+)). No significant bias was observed between CD4.A (+/+) and CD4.B (+/+) PBMCs, although the regression coefficient slightly varied between the DP cells (Fig 4B). Collectively, no significant difference in cellular proportion was observed between CD4.A (+/+) and CD4.B (+/+) PBMCs.

In the next experiment, we stimulated the T cells with superantigen [35] to analyze the activation level of swine CD4 T cells. Because class I SLA expression is enhanced by toxic shock syndrome toxin-1 (TSST-1) stimulation [22, 35], we used the molecule as an activation marker. Also, since SLA polymorphism may affect the reactivity of the anti-class I SLA mAb (X2F6) [23] we selected samples of CD4.A (+/+) and CD4.B (+/+) to analyze the CD4 T cell activation levels in Microminipigs with the homozygous haplotypes Hp-35.23 (CD4.A (+/+); n = 1, CD4.B (+/+); n = 2) and Hp-43.37 (CD4.B (+/+); n = 2). The PBMCs of these homozygous haplotype samples were collected and stimulated *in vitro* with TSST-1; CD4 expression was detected with the prepared mAbs by flow cytometry. To obtain a double-staining pattern, the mAb x1E10 was labeled with biotin (S4 Fig).

After the stimulation of PBMCs with TSST-1, the lymphocyte gate for the activated lymphocytes was enlarged and a cluster of CD4-high cells appeared in the DP cell fraction followed by a decrease of the CD4SP cell ratio (Fig 5 and S5 Fig). In addition, a significant decreases of CD4/CD8 double negative (DN) and CD4SP cells and significant increases of CD8SP and DP cells were observed for the CD4.A (+/+) and the CD4.B (+/+) swine groups (Fig 5B). When the two haplotype groups were examined, both the Hp-35.23 and the Hp-43.37 swine showed a significant increase of the CD4-high fraction after TSST-1 stimulation. Similarly, both the CD4.A (+/+) PBMCs and the CD4.B (+/+) PBMCs showed a significant increase of CD4-high fractions (Fig 5C). Also, the MFI of CD4 on DP cells was increased in the CD4 fraction of the Hp-35.23 haplotype cells (Fig 5D).

We then analyzed the class I SLA expression level of these DP cells. TSST-1 stimulation induced the enhancement of class I SLA protein on a fraction of the CD4+ cells after 72 hrs (S6A Fig). The CD4-high fraction showed a higher expression of class I SLA (S6B Fig), and no difference was observed between the CD4.A (+/+) and CD4.B (+/+) alleles. Similarly, no significant difference was observed between Hp-35.23 and Hp-43.37 (S6C Fig).

To determine whether anti-CD4 mAb treatment could block the CD4 T cell activation, two different samples of PBMCs (#1021 and #3343) were stimulated with TSST-1 in the presence of x1E10 for 72 hrs. As shown in Fig 6 and S7 Fig, the increase in the ratio of CD4-high DP cells in the presence of x1E10 was markedly less than the increase in the ratio of cells cultured

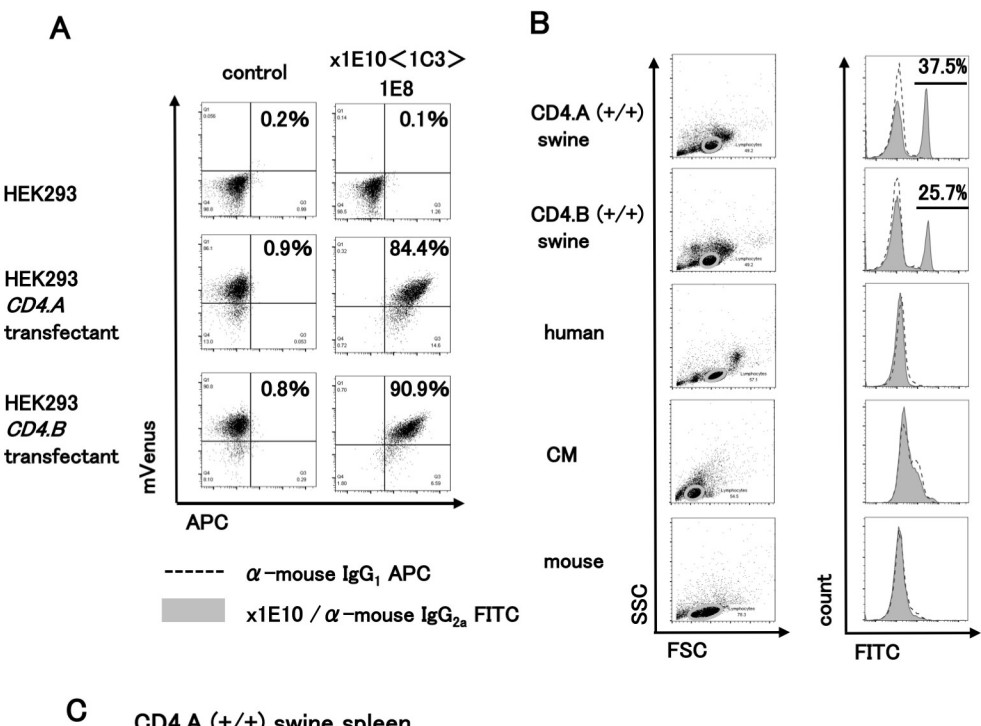

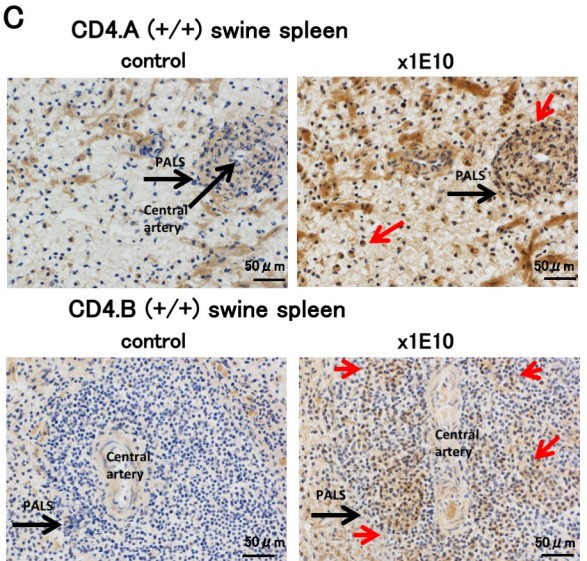

**Fig 3. Characterization of anti-swine CD4.AB mAb (x1E10).** A. HEK293 and the transfectant of *CD4.A* and *CD4.B* were stained with x1E10 (1C3-1E8) clone supernatant and anti-mouse IgG-APC and analyzed by FCM. The numbers in the panels show the percentage of mVenus and APC double-positive cells. B. Swine PBMCs (CD4.A (+/+) and CD4. B (+/+)) were stained with the purified mAb and anti-mouse IgG2a-FITC and analyzed by FCM. The species specificity was examined using CD4.A (+/+) and CD4.B (+/+) swine, human, CM and mouse PBMCs. Lymphocyte gate (orbits) was used for the analysis (left panels). Solid lines and shaded peaks represent the x1E10-stained patterns (right panels). Broken lines represent the isotype-control-stained patterns. The bars in the histograms indicate the positions of x1E10 positive cell peaks together with the percentage of positive cells. C. CD4.A (+/+) (upper panels) and CD4.B (+/+) (lower panels) swine spleen tissue sections stained with x1E10 mAb (right panels) and negative controls (no x1E10 mAb) (left panels). Brack arrow points to PALS and central arteries. Red arrows point to highly cross-reactive cells.

in the absence of x1E10 during the 72 hrs for both swine samples. In comparison to the CD4-high DP cells, the CD4-low DP cells expressed not only CD4 but also class I SLA at lower levels

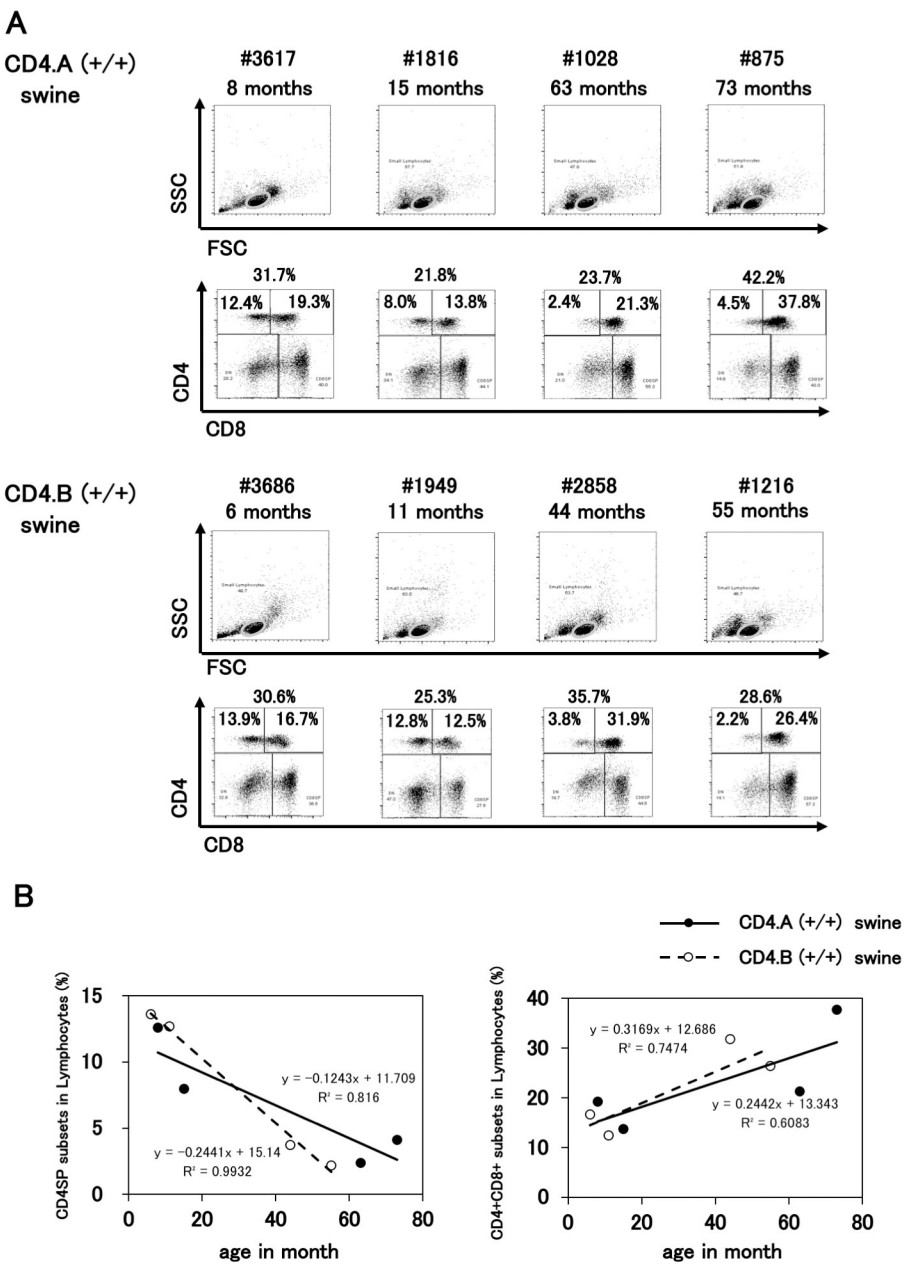

**Fig 4. Surface expression of CD4 and CD8 molecules on the swine T cells.** A. Swine PBMCs (CD4.A (+/+)) were stained with the mAb (x1E10) and anti-CD8 mAb. Small lymphocyte-gated cells were analyzed. Individual identification numbers and age in months are shown on the top of each upper panel. The lower quadrant panel shows the CD4/CD8 expression of the PBMCs. The percentages of CD4SP and DP cells respectively, are shown in the panels. B. Correlation between age in months and CD4SP cell subsets (% in lymphocyte-gated cells) (left panel) and DP cell subsets (% in lymphocyte-gated cells) (right panel) in swine PBMCs. Solid line; CD4.A (+/+) swine, broken line; CD4. B (+/+) swine.

and the cell size was not enlarged with x1E10 treatment. These results suggest that the CD4-low DP cells did not involve activated DP cells and that the CD4-low DP cells could not enhance the surface expression of CD4 because of the anti-CD4 mAb binding and CD4 internalization. However, the ratio of CD4-high DP T cell still increased linearly during 72 hrs

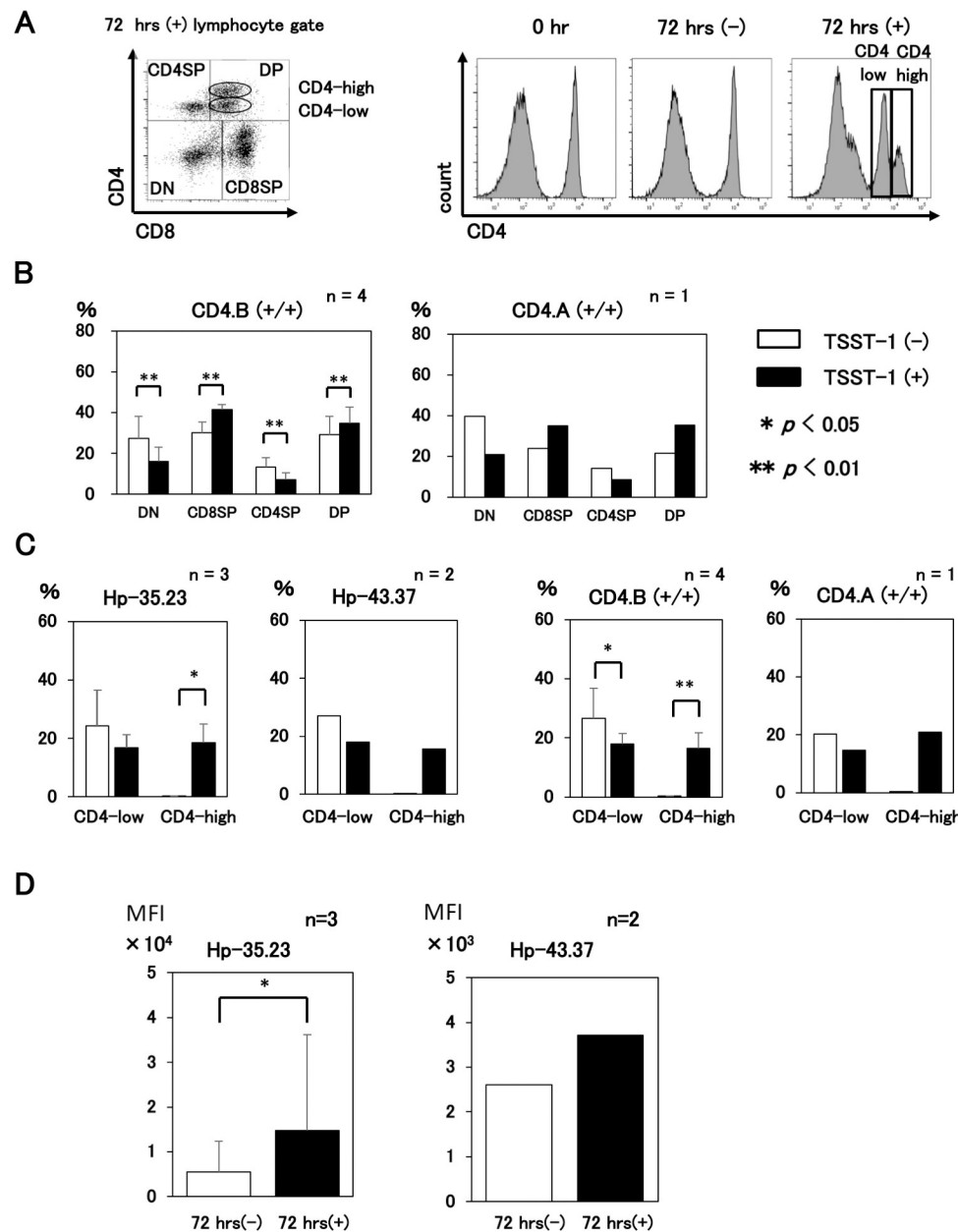

**Fig 5. CD4 expression on DP cells is enhanced by TSST-1 stimulation.** Swine PBMC samples were stained with x1E10 and anti-CD8 mAb and analyzed by FCM. A. The left panel shows the representative pattern of CD4 and CD8 expression in the small and large lymphocyte-gated cells of the DP cells that were divided into CD4-high and CD4-low cells, respectively. The right panels show the histograms of the CD4 expression of the lymphocyte-gated cells. Left panel, 0 hr; middle panel, 72 hrs without TSST-1 stimulation; right panel, 72 hrs with TSST-1 stimulation. B. Four samples of CD4.B (+/+) and one sample of CD4.A (+/+) PBMCs were analyzed for changes in the ratio of T cell subsets with/without TSST-1 stimulation. Statistical analysis was performed on the four samples of the CD4.B (+/+) PBMCs. Open bars, TSST-1 (-); closed bars, TSST-1 (+). C. The changes in the ratio of CD4-low and CD4-high fractions with/without TSST-1 stimulation were compared between Hp-35.23 and Hp-43.37 haplotypes (left two panels) and between the CD4.B (+/+) and CD4.A (+/+) T cells (right two panels). D. MFI of DP cells was compared between samples of TSST-1 (+) and TSST-1 (-) cells. Left panel, three samples of Hp-35.23; right panel, two samples of Hp-43.37. Student's *t*-test was performed for significance in the Hp-35.23 group with the analysis of three samples. * p<0.05. **p<0.01. Open bars; TSST-1 (-), closed bars; TSST-1 (+).

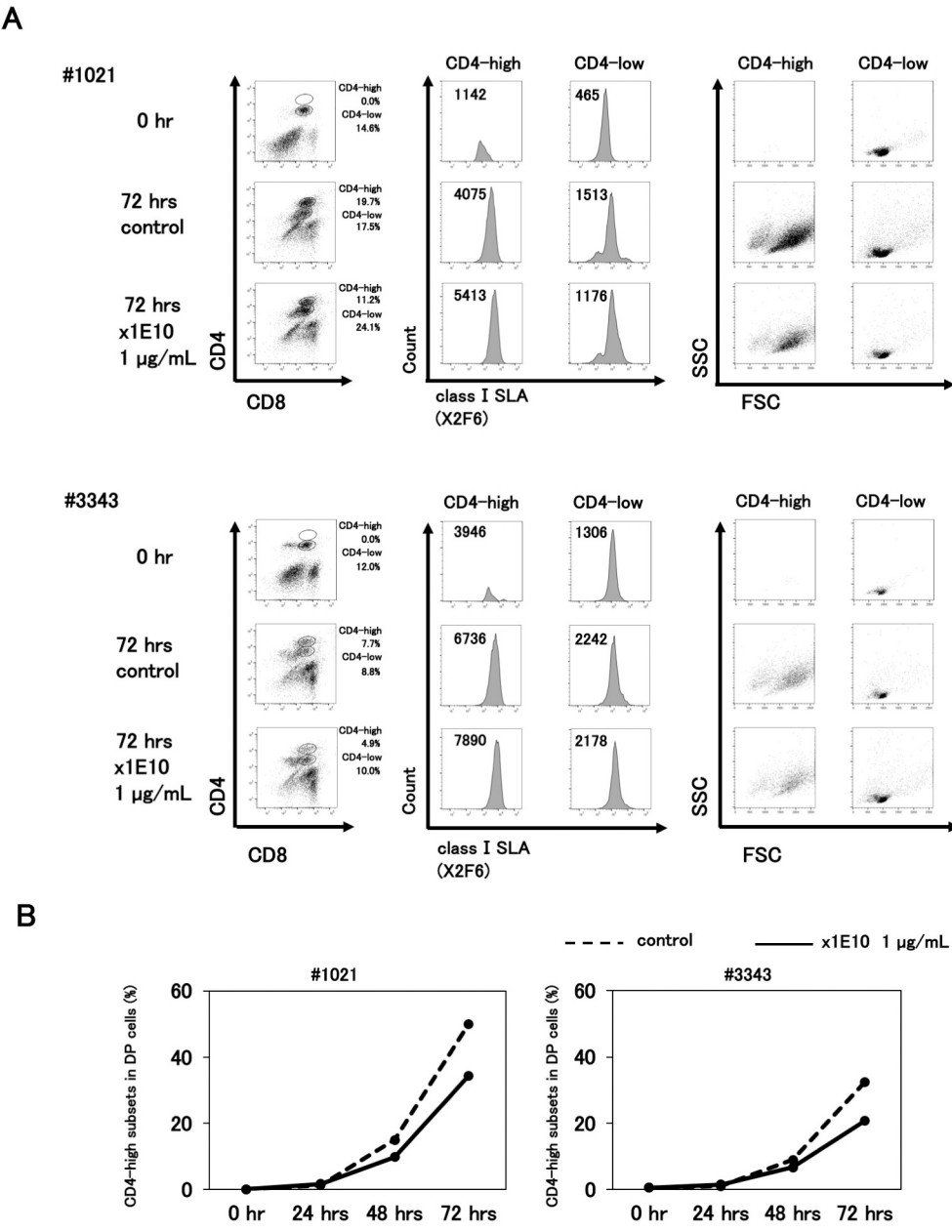

**Fig 6. Activation of CD4 T cells by TSST-1 is inhibited by an anti-CD4 mAb x1E10.** Two swine PBMC samples (upper panels; #1021, lower panels; #3343) were stained with x1E10, anti-CD8 mAb and X2F6, then analyzed through FCM. A. Left panels show the pattern of CD4 and CD8 expression. The DP cells were divided into CD4-high and CD4-low cells. Center panels show the histograms of the class I SLA expression of the CD4-high (left) and CD4-low (right) DP cells. Right panels show the FSC/SSC gates of CD4-high (left) and CD4-low (right) DP cells. Cells were collected at 0 hr (upper panels) and after 72 hrs for negative controls (center panels) and x1E10 (1 μg/mL) mAb treatment (lower panels). B. Kinetics of CD4-high DP cell fraction ratio/DP cell fraction in two TSST-1 stimulated PBMC samples in the presence of x1E10 (1 μg/mL) (solid lines) and absence of x1E10 (broken lines) as a negative control. Left panel, #1021; Right panel, #3343.

incubation, so that the blocking effect of anti-CD4 mAb was only to slightly delay or lower the T cell stimulation by TSST-1 (Fig 6B).

These results with the x1E10 mAb show that both CD4.A and CD4.B bearing T cells can be activated and enhance the expression of CD4 and class I SLA molecules on the surface by superantigen stimulation. Therefore, both CD4.A and CD4.B are functional molecules of the Microminipig. The mAb x1E10 also was shown to inhibit the kinetics of CD4 T cell activation induced by TSST-1.

## Discussion

Although several anti-swine CD4 mAbs were already available commercially, the CD4.B anti-genic variant present in the Microminipig herd did not react with them [17]. The two novel and specific mAbs that we prepared in this study recognized both CD4 isotypes, CD4.A and CD4.B. One of them, b1D7, recognized only CD4.B, whereas the other mAb, x1E10, recognized both of the isotypes. The mAb that recognized both CD4.A and CD4.B was useful for measuring total CD4 expression by FCM and/or immunohistochemistry. These mAbs did not recognize the PBMC of other species when examined by flow cytometry suggesting that they were swine-specific.

The CD4 protein is well-known as one of the receptors that bind to the HIV virus, and the interaction has been detailed at the structural level [36–38]. The molecular structure of CD4 was analyzed in various animal species and comparative analyses revealed that CD4 possesses various isotypes in most of the examined species [9–11, 13, 39]. In the MeLiM minipig strain, the site of mutation and the amino acid substitutions among the CD4 isotypes appear to affect the MHC response [9]. While a report suggested that a human CD4 SNP (C868T) affected the infection of HIV virus, another group reported that no significant difference was observed [13]. Thus, the significance of the effects of CD4 polymorphism on the immune reaction is still unclear.

The CD4 protein is thought to recruit Lck in the intracellular domain when the TCR of T cells and the class II MHC of antigen-presenting cells interact and amplify the activation signal [3]. However, the CD4 molecule may also possess a weak binding affinity to class II MHC in the absence of TCR involvement. Consequently, a hypothesis was proposed that CD4 adheres to the class II MHC-expressing cells and that the CD4 D1 and D2 domains play an important role in the interaction with class II MHC [6]. However, the difference of affinity to class II MHC among the CD4 isotypes has not yet been clarified. To examine the kinetics of protein-protein interaction of CD4 isotypes, mAbs that recognize each allele and/or both alleles are essential. Antibodies that distinguish each isotype CD4 separately may enable the evaluation of the CD4 and class II MHC and/or peptide interaction at the protein level.

Using the new mAbs prepared for this study and the anti-class I SLA mAb that we had already established [23], together with the swine PBMC haplotypes that match the class I SLA allele-specific antibody, we found that the CD4.A and CD4.B proteins are expressed on swine T cells at a similar level. A subset of the CD4 T cells expressed CD8 simultaneously. Stimulation with superantigen TSST-1 induced activation of both CD4.A and CD4.B-expressing T cells, enhanced the expression of both CD8 and CD4, and was followed by the enhancement of specific class I MHC. In our study, the CD4-low DP cells expressed lower levels of class I SLA showing that most of the small memory DP cells are not activated further.

We previously reported that TSST-1 stimulation enhanced the expression of class I SLA on CD4 T cells [22, 23]. In this study, we used this phenomenon to detect the degree of T cell activation. As a result, we observed the increase of CD4, CD8 and class I SLA in TSST-1 stimulated CD4 T cells. Swine PBMCs are reported to involve significant amounts of CD4+CD8+ double-positive (DP) T cells [34]. These DP cells are derived from CD4 single-positive (SP) cells and have a memory phenotype. The reason why these T cells express CD8a has not been clarified

yet. However, the newly expressed CD8 may recognize class I MHC together with the TCR to enhance the binding ability of T cells to antigen-presenting cells (APCs), which express both class I and class II MHCs. Moreover, these activated cells enhance CD4 expression. Therefore, the activated Th cells expressing higher level of CD4 and CD8 might induce stronger signals compared to CD4 SP and CD8 SP cells. Our results show that the activated DP positive cells enhance CD4 and class I SLA expression, which may enhance the TCR signals among the Th cells. Therefore, in this regard, the older animals in swine populations possess such DP cells probably to better support the immune response even if the newly provided immune cells were decreased.

Surprisingly, the anti-CD4 antibody x1E10 suppressed the TSST-1 activation of CD4 T cells and the increase in cell size and class I SLA expression. Because superantigens crosslink TCRβ and MHCβ directly, and CD4 also bind to MHCβ, it might be difficult for the CD4 molecule to be involved in the TSST-1 response. However, the binding of CD4 with the antibody may reduce the number of CD4 molecules engaging in the TCR-MHC interaction. Otherwise, if CD4 functions as an adhesion molecule that directly adheres to the MHC molecules before the cognate interaction, and the interaction is mandatory to construct the cognate interaction, the activation of CD4 T cells might be reduced by inhibiting the CD4-MHC interaction [6–8] with the anti-CD4 antibody.

Taken together, our results suggest that the mAbs prepared in this study have specific advantages because the two new antibodies recognize the CD4.B protein that could not be detected by conventional mAbs. Moreover, the quantitative analysis comparing CD4.A and CD4.B expression is possible because the antibody x1E10 recognizes both CD4.A and CD4.B isotypes. Using both of these novel mAbs and the SLA-stabilized and CD4 isotype-identified Microminipigs in our experiments, we found that CD4.B is expressed at about the same level as CD4.A, which was detected previously using the conventional mAb (PT85A). However, more data need to be collected to determine whether the CD4.A and CD4.B molecules have some functional differences that so far have been undetected. Also, the suppression of the TSST-1 stimulatory effect on T-cell activation by one of our antibodies suggests that taken together both of these new antibodies are promising reagents to support porcine T cell activation studies in future.

## Supporting information

**S1 Fig. Immunization of BALB/c mice with swine CD4 isotype proteins.** A. Protocol for immunization of the BALB/c mice with swine CD4 antigen. Swine PBMC or CD4 isotype transfectants (*CD4A*+/A20 or *CD4B*+/A20) were immunized biweekly. One week after 3rd and 6th immunization, peripheral blood (PB) was collected and antibody titers were examined. B. Antibody reactivity to CD4.A and CD4.B was checked by FCM. The fluorescent intensity of mVenus shows the transfection efficiency. Mouse antisera were used for the first antibody for staining the cells. The numbers shown in the panels are the percentage of the positive cells. Left panels show the CD4.B specific mAb preparation. Right panels show the CD4.A and CD4. B specific mAb preparation.
(TIF)

**S2 Fig. IgH and Igk characterization of b1D7.** The repertoire analysis was performed by Repertoire Genesis (Osaka, Japan). The types of heavy chain and light chains and the sequences are shown.
(TIF)

**S3 Fig. IgH and Igk characterization of x1E10.** The repertoire analysis was performed by Repertoire Genesis (Osaka, Japan). The type of heavy chain and light chains and the sequences

are shown.
(TIF)

**S4 Fig. Evaluation of Biotinylated x1E10 mAb.** The x1E10 mAb was labeled with biotin and HEK293 transfected with *CD4.A* or *CD4.B* gene was stained with biotinylated x1E10 and streptavidin PE-cy7. The reactivity was evaluated by FCM showing that the reactivity of labeled antibody against both alleles is almost the same.
(TIF)

**S5 Fig. Appearance of CD4 high fractions after the stimulation of Hp-35.23 swine PBMC.** Swine PBMCs (CD4.A(+/+) and CD4.B(+/+)) were stimulated and stained with the mAb (x1E10) followed by anti-mouse IgG-PE, and anti-CD8 mAb and analyzed by FCM as mentioned in Fig 5. Left panels; CD4.A (+/+) swine. The same swine collected at a different time (#3617–1 and #3617–2). Right panels; CD4.B (+/+) swine. Two different swine (#2858, #3686) are shown. The ratio of CD4SP and DP cells in the lymphocyte gated cells are shown in the panels. The number shown above each panel represents the sum of CD4SP and DP cell percentages.
(TIF)

**S6 Fig. Class I SLA expression is increased in the TSST-1 stimulated CD4+ cells.** Swine PBMCs were stained with x1E10 and anti-CD8 mAb and analyzed by FCM. A. The left two panels show the representative pattern of FSC/SSC after 72 hrs of culture with/without TSST-1 stimulation. As the large cells are increased by TSST-1 stimulation, a small and large lymphocyte gate was used for the analysis. CD4/CD8 expression is shown in middle panels. Upper panel shows the CD4/CD8 pattern of Hp-35.23 and lower panels; Hp-43.37. Right panels show the histograms of the class I SLA expression of the lymphocyte-gated T cell subsets after TSST-1 stimulation. X2F6 was used for Hp-35.23 and PT85A was used for Hp-43.37. B. The left panel shows the gate of each CD4+ fraction analyzed for the expression of class I SLA. The CD4/CD8 DP cells were divided into CD4 high and CD4 low groups to examine the expression levels of class I SLA and the MFI data are shown in the right table. The middle panels show the overlay pattern of class I SLA expression in each group of Hp-35.23 and Hp-43.37 swine. The groups were further divided into CD4.A(+/+) and CD4.B(+/+) groups that are shown in the panels. C. Class I SLA expression on T cells after TSST-1 stimulation. Left panel; The MFIs of class I SLA expression on DP T cells of Hp-35.23 swine with/without stimulation of TSST-1. Right panel; class I SLA expression on DP T cells of Hp-43.37. Open bars; 72 hrs culture without TSST-1, Black bars; 72 hrs culture with TSST-1.
(TIF)

**S7 Fig. Appearance of CD4 high fractions after the stimulation of swine PBMC.** Swine PBMCs (#1021 and #3343) were stimulated in the absence and presence of x1E10. Samples were stained with the mAb (x1E10) followed by anti-mouse IgG-PE. Then, the cells were stained with anti-CD8 mAb and analyzed by FCM as described in Fig 5. Left 4 panels, without TSST-1 stimulation; Right 4 panels, with TSST-1 stimulation. Control groups are stained without x1E10. The x1E10 groups were cultured in the presence of x1E10. The samples were collected at 24, 48 and 72 hrs after the stimulation. The cell size and shape were measured by forward scatter (FSC) and side scatter (SSC). The expressions of CD4 and CD8 were detected in the lymphocyte-gated cells. The data presented in the Figure was used for the Fig 6B.
(TIF)

**S1 Table.**
(PDF)

**S2 Table.**
(PDF)

**S3 Table.**
(PDF)

**S4 Table.**
(PDF)

## Acknowledgments

We thank Fuji Micra Inc. (Fujinomiya, Japan) for providing blood samples of Microminipigs. We thank the members of the Teaching and Research Support Center in the Tokai University School of Medicine for their technical skills.

## Author Contributions

**Conceptualization:** Noriaki Imaeda, Masaki Takasu, Takashi Shiina, Hitoshi Kitagawa, Asako Ando, Yoshie Kametani.

**Data curation:** Shino Ohshima, Tatsuya Matsubara, Asuka Miyamoto, Atsuko Shigenari, Yoshie Kametani.

**Formal analysis:** Shino Ohshima, Shingo Suzuki, Noriaki Hirayama, Yoshie Kametani.

**Investigation:** Shino Ohshima, Tatsuya Matsubara, Asuka Miyamoto, Atsuko Shigenari, Masafumi Tanaka, Shingo Suzuki, Asako Ando, Yoshie Kametani.

**Methodology:** Noriaki Hirayama, Asako Ando, Yoshie Kametani.

**Project administration:** Asako Ando, Yoshie Kametani.

**Resources:** Tatsuya Matsubara, Noriaki Imaeda, Masaki Takasu, Masafumi Tanaka.

**Software:** Takashi Shiina, Shingo Suzuki, Noriaki Hirayama.

**Supervision:** Takashi Shiina, Hitoshi Kitagawa, Asako Ando, Yoshie Kametani.

**Validation:** Shino Ohshima, Asuka Miyamoto, Atsuko Shigenari, Masaki Takasu.

**Visualization:** Shino Ohshima, Yoshie Kametani.

**Writing – original draft:** Shino Ohshima, Yoshie Kametani.

**Writing – review & editing:** Hitoshi Kitagawa, Jerzy K. Kulski, Yoshie Kametani.

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
