## [Decision Letter · Decision Letter 0]

22 Jun 2020

PONE-D-20-14895

Preparation and characterization of monoclonal antibodies recognizing two CD4 isotypes of Microminipigs

PLOS ONE

Dear Dr. Kametani,

Thank you for submitting your manuscript to PLOS ONE. After careful consideration, we feel that it has merit but does not fully meet PLOS ONE’s publication criteria as it currently stands. Therefore, we invite you to submit a revised version of the manuscript that addresses the points raised during the review process.

As you will see, the reviewers expressed generally favorable views of the manuscript in the context of PLOS ONE publication criteria, but both also suggested important revisions. I ask that you particularly focus on the recommendations to further support the utility of the new anti-CD4 monoclonal antibodies in characterizing the superantigen responses (Fig 5); revisit the proposed supplemental figures with a view toward selecting only those that enhance the results in the main body of the manuscript; and, remembering that PLOS ONE does not copyedit manuscripts, please edit the text carefully throughout for clarity and brevity (the reviewers have provided some specific editorial suggestions for your consideration).

We look forward to receiving your revised manuscript.

Kind regards,

Karen L. Elkins

Academic Editor

PLOS ONE

Journal Requirements:

Additional Editor Comments (if provided):

Reviewers' comments:

Reviewer's Responses to Questions

**Comments to the Author**

1. Is the manuscript technically sound, and do the data support the conclusions?

Reviewer #1: Yes

Reviewer #2: Yes

2. Has the statistical analysis been performed appropriately and rigorously? 

Reviewer #1: Yes

Reviewer #2: Yes

3. Have the authors made all data underlying the findings in their manuscript fully available?

Reviewer #1: Yes

Reviewer #2: Yes

4. Is the manuscript presented in an intelligible fashion and written in standard English?

Reviewer #1: Yes

Reviewer #2: No

5. Review Comments to the Author

Reviewer #1: In their manuscript entitled “Preparation and characterization of monoclonal antibodies recognizing two CD4 isotypes of Microminipigs” Shino Ohshima et al. describe the development of monoclonal antibodies against gene products of CD4.A and CD4.B alleles in swine.  In this manuscript the authors focus on the CD4.B expression. Interestingly the gene product of CD4.B is not recognized by most of the mAb used for the characterization of porcine CD4+ T cells and there is indeed a lack of reagents.

This is an interesting manuscript and the molecular data are highly appreciated. In regard to the functional studies with the superantigen stimulation I have some questions and concerns.

1. Can the authors block their superantigen stimulation with their anti-CD4 mAb?

2. Can one block a superantigen stimulation with anti CD4 mAb at all?

3. Is CD4 at all involved in the superantigen stimulation? I do not think so.

4. To get a more precise information I would strongly recommend to analyze the function of the respective CD4 molecules in an antigen-specific immune response e.g. against a vaccine antigen or ovalbumin.

5. Summerfield et al. (Cellular Immunology 168, 291-296 (1996)) could show a blocking of their PRV-specific T-cell response with mAb 74-12-4. Would the new mAbs do the same?

6. An alternative might be to try blocking of a CD4 response in allogeneic mixed leukocyte cultures with the novel antibodies.

Additional points:

7. Lines 95 -100 these sentences are hard to understand. Why do the authors use anti-MHC-I antibodies?

8. Line 305: introduced

9. Does the novel anti-CD4 mAb against CD4.A (x1E10) recognize another epitope than the existing mAbs e.g. 74-12-4?

10. Discussion line 531- 541: What is the deeper sense to discuss MHC-I expression in a CD4 manuscript? I think this part of the discussion should be deleted.

11. These cells have an activated and memory phenotype. Can the authors give any information about central and effector memory cells?

12. Line 554 – 566: Personally, I do not think that this part of the discussion contributes a lot to the CD4 story.

13. The immunohistochemistry presented in figure 3c is poor. It would have been of advantage to see e.g. in lower magnification in spleen staining of the PALS or in lymph node staining of T-cell areas. The same for thymus. A comparison of the respective staining pattern with staining pattern of existing anti-CD4 mAb would have been helpful.

14. Figure 5: is there a reason why the authors selected after 72h stimulation with the superantigen a lymphocyte gate? Would it not be much better to analyze the stimulated cells in a blast gate as presented in figure 6 and compare their phenotype with the non-stimulated resting cells in the lymphocyte gate. Another way to get the phenotype of activated proliferating cells would be the use of CFSE or violet and a respective gating on proliferating cells. These additional analyses would enhance the quality of this manuscript significantly.

15. Figure 6 should be skipped or moved to supplementary figures

16. Figure S6 should be skipped and the results mentioned in the text

17. The same for figure S7.

Reviewer #2: This manuscript is on generation of novel monoclonal antibody reagents valuable for characterization of CD4 T cell subpopulations including CD4.A and CD4.B alleles in minipigs with different haplotypes. The authors also tried to use two developed antibody clones recognizing CD4.A and CD4.AB specifically in characterizing the up-regulation of CD4 molecule and SLA on T-cells in PBMC stimulated with TSST-1. The analysis using these monoclonal antibodies could define the specific population of CD4.B T cells activated after TSST-1 stimulation. However, a major revision on following aspects should be made before further review and consideration for publication in PLOS One journal.

Major comments:

1. Overall English writing and grammar should improved for clear communication to readers. It is difficult to understand the authors' interpretation and meaning.

2. Lines 372-488 are not clearly written and have many grammatical errors and incomplete sentences. It is difficult to understand authors' meaning. Rewrite the section on comparison of ClassI SLA expression on T cells with different CD4 isotypes. Discussion sentences in this result section should be moved to Discussion part.

3. Lines 459-462 and fig 6C used two different antibodies for two pigs with different haplotypes to analyze the sensitivity to TSST-1 stimulation. How can you interpret whether the difference in sensitivity is due to pig haplotypes or monoclonal antibodies use?

4. Lines 485-486:

Minor comments:

1. Lines 39: mRNA expression needs to be changed into mRNA transcription.

2. Line 202: The sequences of monoclonal antibody hybridoma clones should be submitted to GenBank and submission numbers need to be included in the text.

3. Line 259: Replace 'was' with 'were'.

4. Line 280: Replace 'become' with 'be'.

5. Line 290: Replace 'is' with 'was'.

6. Line 298: Delete 'in Microminipigs each'.

7. Line 306 and many other sentences: Replace 'cross-reactivity' with 'specific reactivity' or 'reactivity'.

8. Line 309: Change a peak of mean into a peak mean.

9. Lines 313-314: 'Produced both---- specific antibodies.' should be changed into 'both the CD4.B and the CD4.A specific antibodies were produced.'

10. Line 333 figure 2 and other figures: 'mVenus' channel needs to be described.

11. Lines 345 and 346: Change 'recognizes' into 'recognized' for use of identical tense in the sentence.

12. Line 346: Delete 'also'.

13. Line 347: Change 'PBMC but only' into PBMC, but recognized only.

14. Line 348: Change 'Accordingly' into 'Therefore'.

15. Lines 387 and 388: Delete 'were this' and 'detected'. move 'also' right after but to make 'but also'.

16. Line 399: 'Individual numbers' should be changed into 'Individual identification numbers'.

17. Line 402: Change 'Correlation of' into 'Correlation between'.

18. Many more English errors need to be fixed.

6. PLOS authors have the option to publish the peer review history of their article (what does this mean?). If published, this will include your full peer review and any attached files.

Reviewer #1: Yes: Armin Saalmüller

Reviewer #2: No

---

## [Author Response · Author response to Decision Letter 0]

10 Sep 2020

Reviewer #1: In their manuscript entitled “Preparation and characterization of monoclonal antibodies recognizing two CD4 isotypes of Microminipigs” Shino Ohshima et al. describe the development of monoclonal antibodies against gene products of CD4.A and CD4.B alleles in swine. In this manuscript the authors focus on the CD4.B expression. Interestingly the gene product of CD4.B is not recognized by most of the mAb used for the characterization of porcine CD4+ T cells and there is indeed a lack of reagents.

This is an interesting manuscript and the molecular data are highly appreciated. In regard to the functional studies with the superantigen stimulation I have some questions and concerns.

1. Can the authors block their superantigen stimulation with their anti-CD4 mAb?

2. Can one block a superantigen stimulation with anti CD4 mAb at all? 

3. Is CD4 at all involved in the superantigen stimulation? I do not think so.

4. To get a more precise information I would strongly recommend to analyze the function of the respective CD4 molecules in an antigen-specific immune response e.g. against a vaccine antigen or ovalbumin.

5. Summerfield et al. (Cellular Immunology 168, 291-296 (1996)) could show a blocking of their PRV-specific T-cell response with mAb 74-12-4. Would the new mAbs do the same?

6. An alternative might be to try blocking of a CD4 response in allogeneic mixed leukocyte cultures with the novel antibodies.

We appreciate the reviewer to give us important suggestions. Till now, we cannot find any study that describe the blocking of superantigen stimulation with anti-CD4 mAb. Because the superantigen reaction only use TCRβ and MHC class II, CD4 molecules were supposed not to be involved. Therefore, we did not check the function in the previous version. According to the reviewer’s suggestions, we performed an experiment for blocking of TCR signals. Because the vaccine response and allogeneic mixed leukocyte reaction (MLR) are different among SLA-haplotypes because the epitope peptide-presentation or allogeneic MHC recognition depend on the MHC type, we performed TSST-1 stimulation but not vaccine stimulation. As a result, our mAb x1E10 delayed the activation of CD4 T cells when they were added to TSST-1 stimulation. We showed the results in Fig.6. It is not clear why TSST-1 stimulation could be suppressed by anti-CD4 mAb, and the issue will be addressed in near future. While Microminipig herds have not been bred to select SLA homozygous pigs, we will try to select the most reactive combination of the haplotypes and perform the experiments in future.

Additional points:

7. Lines 95 -100 these sentences are hard to understand. Why do the authors use anti-MHC-I antibodies?

According to the reviewer’s comment, we changed the description of Lines 94~99 as follows, 

“In this study, we developed two mAbs which recognize Microminipig CD4.B, one for only CD4.B and the other for CD4.A and CD4.B in order to analyze CD4.A and CD4.B protein expression and function. We used these antibodies to evaluate (1) the level of CD4 protein expression and T cell activation, and (2) the association of T cell activation with class I SLA protein expression levels in response to in vitro T cell stimulation with the toxic shock syndrome-1 (TSST-1) enterotoxin.” 

8. Line 305: introduced

We changed the description and the word ‘introduced was not used in the sentence of Lines 318~320.

9. Does the novel anti-CD4 mAb against CD4.A (x1E10) recognize another epitope than the existing mAbs e.g. 74-12-4?

We think that the antibody (x1E10) recognizes another epitope, where CD4.A and CD4.B share common sequence, although the epitope of x1E10 has not been defined yet,

10. Discussion line 531- 541: What is the deeper sense to discuss MHC-I expression in a CD4 manuscript? I think this part of the discussion should be deleted.

As reviewer’s suggestion, we deleted the description and concentrated on the CD4 function.

11. These cells have an activated and memory phenotype. Can the authors give any information about central and effector memory cells?

We appreciate the reviewer’s opinion. We don’t have any data supporting the evidences till now. However, as it is very important, we will check the phenotype in future.

12. Line 554 – 566: Personally, I do not think that this part of the discussion contributes a lot to the CD4 story.

According to the reviewer’s suggestion, we deleted the part.

13. The immunohistochemistry presented in figure 3c is poor. It would have been of advantage to see e.g. in lower magnification in spleen staining of the PALS or in lymph node staining of T-cell areas. The same for thymus. A comparison of the respective staining pattern with staining pattern of existing anti-CD4 mAb would have been helpful.

According to the reviewer’s suggestions, we added a lower magnification image (x10), although we could not obtain the thymus tissues. As shown in the Fig 3, PALS and central arteries are clearly shown. Brown signals are involved in the PALS, suggesting the CD4 T cell accumulation.

14. Figure 5: is there a reason why the authors selected after 72h stimulation with the superantigen a lymphocyte gate? Would it not be much better to analyze the stimulated cells in a blast gate as presented in figure 6 and compare their phenotype with the non-stimulated resting cells in the lymphocyte gate. Another way to get the phenotype of activated proliferating cells would be the use of CFSE or violet and a respective gating on proliferating cells. These additional analyses would enhance the quality of this manuscript significantly.

We already analyzed the activation time and found that after 72 hrs the activation of Th cells are highly detectable (ref 22). During 72 hrs of the activation, cells become enlarged and increase expression. According to the reviewer’s suggestion, we added the time course of CD4 and expression from 0 to 72 hrs in the revised version (Fig 6B, S7 Fig).

15. Figure 6 should be skipped or moved to supplementary figures

According to the reviewer’s suggestion, we moved Fig. 6 to supplemental data ( S6 Fig).

16. Figure S6 should be skipped and the results mentioned in the text

17. The same for figure S7.

According to the reviewer’s suggestion, we deleted them.

 

Reviewer #2: This manuscript is on generation of novel monoclonal antibody reagents valuable for characterization of CD4 T cell subpopulations including CD4.A and CD4.B alleles in minipigs with different haplotypes. The authors also tried to use two developed antibody clones recognizing CD4.A and CD4.AB specifically in characterizing the up-regulation of CD4 molecule and SLA on T-cells in PBMC stimulated with TSST-1. The analysis using these monoclonal antibodies could define the specific population of CD4.B T cells activated after TSST-1 stimulation. However, a major revision on following aspects should be made before further review and consideration for publication in PLOS One journal.

Major comments:

1. Overall English writing and grammar should improved for clear communication to readers. It is difficult to understand the authors' interpretation and meaning.

According to the reviewer’s comment, we extensively improved the English.

2. Lines 372-488 are not clearly written and have many grammatical errors and incomplete sentences. It is difficult to understand authors' meaning. Rewrite the section on comparison of class I SLA expression on T cells with different CD4 isotypes. Discussion sentences in this result section should be moved to Discussion part.

According to the reviewer’s comment, we rewrote the part with improved the English. The section was named as “Comparison of class I SLA expression on T cells with different CD4 isotypes” and emphasized that class I SLA was used as an activation marker.

3. Lines 459-462 and fig 6C used two different antibodies for two pigs with different haplotypes to analyze the sensitivity to TSST-1 stimulation. How can you interpret whether the difference in sensitivity is due to pig haplotypes or monoclonal antibodies use?

As the reviewer pointed out, it was difficult to interpret the difference. In our manuscript, even if the CD4 isotype was different, as shown in S6 Fig B, the CD4 high DP cells showed higher expression of class I SLA (Hp.35.23) compared with CD4 low DP cells. Similar result was obtained using Hp.43.37. However, the case was not enough and it takes time to definitely say that the reaction is same or not. Therefore, we deleted most of the description in the section. Only the increase of class I SLA after the stimulation was mentioned here, which was used to identify the activation of CD4 T cells. 

4. Lines 485-486:

Do you mean the sentence “Therefore, both CD4.A and CD4.B are functional molecules of the Microminipig.” The description itself might be OK.

Minor comments:

1. Lines 39: mRNA expression needs to be changed into mRNA transcription.

According to the reviewer’s suggestion, we changed the sentence and deleted the word.

2. Line 202: The sequences of monoclonal antibody hybridoma clones should be submitted to GenBank and submission numbers need to be included in the text.

Because the mAb was submitted for the patent, the sequence will be automatically sent to GenBank.

3. Line 259: Replace 'was' with 'were'.

According to the reviewer’s suggestion, we changed the word (Line 274).

4. Line 280: Replace 'become' with 'be'.

According to the reviewer’s suggestion, we changed the sentence and deleted the word (Lines 295~299).

5. Line 290: Replace 'is' with 'was'.

According to the reviewer’s suggestion, we replaced the word (Line 307).

6. Line 298: Delete 'in Microminipigs each'.

According to the reviewer’s suggestion, we changed the sentence and deleted the word (Lines 312~314).

7. Line 306 and many other sentences: Replace 'cross-reactivity' with 'specific reactivity' or 'reactivity'.

According to the reviewer’s suggestion, we replaced the word (Line 320).

8. Line 309: Change a peak of mean into a peak mean.

According to the reviewer’s suggestion, we changed the sentence and deleted the word (Line 322~323).

9. Lines 313-314: 'Produced both---- specific antibodies.' should be changed into 'both the CD4.B and the CD4.A specific antibodies were produced.'

According to the reviewer’s suggestion, we changed the description (Lines 325 and 326).

10. Line 333 figure 2 and other figures: 'mVenus' channel needs to be described.

We described about mVenus in materials methods Page 10 Line148-149 as “Transgene-positive cell ratio was determined according to the fluorescent intensity of the co-expressed mVenus protein,” and in Fig2 legend Page 21 Line 345 “The numbers in the panels show the percentage of mVenus and APC double positive cells.” And Fig3 legend Page 22 Line 370-371. The numbers in the panels show the percentage of mVenus and APC double positive cells.”

11. Lines 345 and 346: Change 'recognizes' into 'recognized' for use of identical tense in the sentence.

According to the reviewer’s suggestion, we changed the word (Line 353).

12. Line 346: Delete 'also'.

According to the reviewer’s suggestion, we deleted the word and changed the description.

13. Line 347: Change 'PBMC but only' into PBMC, but recognized only.

According to the reviewer’s suggestion, we changed the word and changed the description.

14. Line 348: Change 'Accordingly' into 'Therefore'.

We changed the description largely and the description was deleted. We described the summarized sentence in Line 354-356.

15. Lines 387 and 388: Delete 'were this' and 'detected'. move 'also' right after but to make 'but also'.

According to the reviewer’s suggestion, we changed the word (Lines 390 to 392).

16. Line 399: 'Individual numbers' should be changed into 'Individual identification numbers'.

According to the reviewer’s suggestion, we changed the word (Line 404).

17. Line 402: Change 'Correlation of' into 'Correlation between'. 

According to the reviewer’s suggestion, we changed the word (Line 407).

18. Many more English errors need to be fixed.

 We checked and corrected the grammar.

---

## [Decision Letter · Decision Letter 1]

21 Oct 2020

PONE-D-20-14895R1

Preparation and characterization of monoclonal antibodies recognizing two CD4 isotypes of Microminipigs

PLOS ONE

Dear Dr. Kametani,

Thank you for submitting your revised manuscript to PLOS ONE, which the reviewers find to have satisfactorily addressed most of the original comments. Pending remaining editorial revisions that are outlined by the reviewers, I believe that the manuscript will meet PLOS ONE’s publication criteria. Please provide a revised manuscript that addresses the remaining points raised during the review process, as outlined below, for a final editorial review.

We look forward to receiving your revised manuscript.

Kind regards,

Karen L. Elkins

Academic Editor

PLOS ONE

Reviewers' comments:

Reviewer's Responses to Questions

**Comments to the Author**

1. If the authors have adequately addressed your comments raised in a previous round of review and you feel that this manuscript is now acceptable for publication, you may indicate that here to bypass the “Comments to the Author” section, enter your conflict of interest statement in the “Confidential to Editor” section, and submit your "Accept" recommendation.

Reviewer #1: All comments have been addressed

Reviewer #2: (No Response)

2. Is the manuscript technically sound, and do the data support the conclusions?

Reviewer #1: Yes

Reviewer #2: Yes

3. Has the statistical analysis been performed appropriately and rigorously? 

Reviewer #1: Yes

Reviewer #2: N/A

4. Have the authors made all data underlying the findings in their manuscript fully available?

Reviewer #1: Yes

Reviewer #2: Yes

5. Is the manuscript presented in an intelligible fashion and written in standard English?

Reviewer #1: Yes

Reviewer #2: Yes

6. Review Comments to the Author

Reviewer #1: In my opinion this manuscript is fine now. The authors corrected most of the topics. Therer are still some open points but as the authors ,mentioned these can be analyzed in further studies.

Reviewer #2: This revised manuscript has significantly improved writing overall. However, one more improvement in grammar and writing described below will make the manuscript more accessible to readers.

Major comment:

1. Lines 48-56 are about practical applications of developed monoclonal antibodies. Please try to rewrite this paragraph and ask your peers to read with fresh eyes.

Minor comments:

2. "Using" would be a more reasonable and concise expression than "By using" in many sentences.

3. Line 46: Using these two mAbs would be more readable than "By using both of these mAbs".

4. Line 46: CD4.B allele-specific or unique regions would read better than "CD4.B allele proteins".

7. PLOS authors have the option to publish the peer review history of their article (what does this mean?). If published, this will include your full peer review and any attached files.

Reviewer #1: **Yes: **Armin Saalmüller

Reviewer #2: No

---

## [Author Response · Author response to Decision Letter 1]

30 Oct 2020

On behalf of all authors,

Reviewer #1: In my opinion this manuscript is fine now. The authors corrected most of the topics. Therer are still some open points but as the authors ,mentioned these can be analyzed in further studies.

Reviewer #2: This revised manuscript has significantly improved writing overall. However, one more improvement in grammar and writing described below will make the manuscript more accessible to readers.

Major comment:

1. Lines 48-56 are about practical applications of developed monoclonal antibodies. Please try to rewrite this paragraph and ask your peers to read with fresh eyes.

We rewrote the lines 48-56 to read as follows:

“Moreover, stimulation of peripheral blood mononuclear cells (PBMCs) derived from CD4.A (+/+) and CD4.B (+/+) swine with toxic shock syndrome toxin-1 (TSST-1) in vitro similarly activated both groups of cells that exhibited a slight increase in the CD4/CD8 double positive (DP) cell ratio. A large portion of the DP cells from the allelic CD4.A (+/+) and CD4.B (+/+) groups enhanced the total CD4 and class I swine leukocyte antigen (SLA) expression. The x1E10 mAb delayed and reduced the TSST-1-induced activation of CD4 T cells. Thus, CD4.B appears to be a functional protein whose expression on activated T cells is analogous to CD4.A.”

Minor comments:

2. "Using" would be a more reasonable and concise expression than "By using" in many sentences.

We removed all ‘by’ from ‘by using’.

3. Line 46: Using these two mAbs would be more readable than "By using both of these mAbs".

We corrected according to the reviewer’s suggestion.

4. Line 46: CD4.B allele-specific or unique regions would read better than "CD4.B allele proteins".

We corrected according to the reviewer’s suggestion.

---

## [Editor Report · Decision Letter 2]

5 Nov 2020

Preparation and characterization of monoclonal antibodies recognizing two CD4 isotypes of Microminipigs

PONE-D-20-14895R2

Dear Dr. Kametani,

We’re pleased to inform you that your manuscript has been judged scientifically suitable for publication and will be formally accepted for publication once it meets all outstanding technical requirements.

Kind regards,

Karen L. Elkins

Academic Editor

PLOS ONE
---

## [Editor Report · Acceptance letter]

13 Nov 2020

PONE-D-20-14895R2 

Preparation and characterization of monoclonal antibodies recognizing two CD4 isotypes of Microminipigs 

Dear Dr. Kametani:

I'm pleased to inform you that your manuscript has been deemed suitable for publication in PLOS ONE. Congratulations! Your manuscript is now with our production department. 

Kind regards, 

on behalf of

Dr. Karen L. Elkins 

Academic Editor

PLOS ONE